# The microRNA-202 as a Diagnostic Biomarker and a Potential Tumor Suppressor

**DOI:** 10.3390/ijms23115870

**Published:** 2022-05-24

**Authors:** Emad A. Ahmed, Peramaiyan Rajendran, Harry Scherthan

**Affiliations:** 1Biological Sciences Department, College of Science, King Faisal University, Hofuf 31982, Saudi Arabia; prajendran@kfu.edu.sa; 2Laboratory of Molecular Physiology, Zoology Department, Faculty of Science, Assiut University, Assiut 71515, Egypt; 3Department of Biochemistry, Saveetha Dental College, Saveetha Institute of Medical and Technical Sciences, Saveetha University, Chennai 600077, India; 4Institut für Radiobiologie der Bundeswehr in Verb. mit der Universität Ulm, 80937 Munich, Germany; scherth@rhrk.uni-kl.de

**Keywords:** microRNA-202, cancer, tumor suppressor, lncRNA, diagnostic biomarkers

## Abstract

MicroRNA-202 (miR-202) is a member of the highly conserved let-7 family that was discovered in *Caenorhabditis elegans* and recently reported to be involved in cell differentiation and tumor biology. In humans, miR-202 was initially identified in the testis where it was suggested to play a role in spermatogenesis. Subsequent research showed that miR-202 is one of the micro-RNAs that are dysregulated in different types of cancer. During the last decade, a large number of investigations has fortified a role for miR-202 in cancer. However, its functions can be double-edged, depending on context they may be tumor suppressive or oncogenic. In this review, we highlight miR-202 as a potential diagnostic biomarker and as a suppressor of tumorigenesis and metastasis in several types of tumors. We link miR-202 expression levels in tumor types to its involved upstream and downstream signaling molecules and highlight its potential roles in carcinogenesis. Three well-known upstream long non-coding-RNAs (lncRNAs); MALAT1, NORAD, and NEAT1 target miR-202 and inhibit its tumor suppressive function thus fueling cancer progression. Studies on the downstream targets of miR-202 revealed PTEN, AKT, and various oncogenes such as metadherin (*MTDH*), *MYCN*, *Forkhead box protein R2* (*FOXR*2) and Kirsten rat sarcoma virus (*KRAS*). Interestingly, an upregulated level of miR-202 was shown by most of the studies that estimated its expression level in blood or serum of cancer patients, especially in breast cancer. Reduced expression levels of miR-202 in tumor tissues were found to be associated with progression of different types of cancer. It seems likely that miR-202 is embedded in a complex regulatory network related to the nature and the sensitivity of the tumor type and therapeutic (pre)treatments. Its variable roles in tumorigenesis are mediated in part thought its oncogene effectors. However, the currently available data suggest that the involved signaling pathways determine the anti- or pro-tumorigenic outcomes of miR-202’s dysregulation and its value as a diagnostic biomarker.

## 1. Introduction

MicroRNAs (miRNAs) are usually short (~22 nucleotides in length), non-coding RNAs involved in posttranscriptional regulation of gene expression or silencing. More than 28,000 miRNAs were reported to be involved in posttranscriptional regulation during physiological and pathological processes, including cell proliferation, carcinogenesis, apoptosis, and angiogenesis [1,2]. While miRNAs play roles in the regulation of apoptosis and suppression of cancer, their altered expression may be associated with the hallmark of tumorigenesis, like activation, invasion, and metastasis, as well as the promotion of proliferation and tumor growth and inhibition of apoptosis [3].

miR-202 is a highly conserved miRNA among vertebrates, including humans and rodents. In humans, it maps to chromosome 10 with its mature single-stranded sequence being 5′-UUCCUAUGCAUAUACUUCUUUG-3′. The mature miR-202 generated from the 5′ and 3′ arms of the pre-miRNA precursor display similar properties and functions [4]. Since nearly 100 reports (as revealed by medical data base search with miR-202 and cancer as mesh terms at www.ncbi.nlm.nih.gov/pmc/, as accessed 29 March 2022) indicate dysregulated expression levels of miR-202 that have been implicated in cancer suppression or promotion, we below review its roles in tumorigenesis in different tissue types.

## 2. Circulating miR-202 as a Diagnostic Biomarker in Cancer Patients

Upregulated miR-202 levels in patient blood has been suggested to be a novel biomarker for early detection of different types of cancer including multiple myeloma (MM) [5], ovarian cancer [6], gallbladder cancer [7], and lung cancer [8]. In MM patients, the relative expression of serum miR-202 was significantly higher than that in healthy controls, suggesting a potential value of miR-202 as a marker for the diagnosis of this tumor [5]. In addition, miR-202 was reported to be among a set of miRNAs that were confirmed to be higher in pre-operative sera of ovarian clear cell carcinoma patients [6]. However, a study comprising 40 samples of gallbladder cancer patients without preoperative treatments revealed significantly increased miR-202 levels in blood and tumor tissues, leading to the suggestion that upregulated miR-202 levels in blood plasma to be a biomarker for early detection of gallbladder cancer [7]. Along that line, lung cancer patients with high plasma miR-202 levels had a higher rate of disease progression and poor prognosis compared to those with low miR-202 expression [8]. Recent clinical studies evaluated the potential diagnostic value of miR-202 expression levels in patient blood at different stages of breast cancer (BC). For instance, in a miRNA profiling study on whole blood of 48 early-stage BC patients, miR-202 was found to be significantly up-regulated based on microarray and RT-qPCR analysis [9]. In addition, circulating miR-202 levels were reported to increase significantly in plasma of BC patients (at different stages) compared to healthy individuals. Moreover, a higher level of serum miR-202 in 34 BC patients was found to be significantly correlated with poor overall survival [10]. On the other hand, in early-stage BC patients, a gradual decrease in miR-202 blood expression level was found to correlate with progression from early stage to tumor-node-metastasis (TNM) stage [11]. Similarly, a study on oral cancer reported decreased miR-202 levels in patient sera and cancer tissues [12].

Together, most of the above studies suggest that the increase in circulating miR-202 may be useful as a diagnostic clinical biomarker for progressed cancer. However, more studies on early to late stage transitions are required to further improve the understanding of the prognostic value of reduced miR-202 levels in peripheral blood, given that few studies observed a decrease of miR-202 blood/sera levels with tumor progression.

## 3. Upregulation of miR-202 in Breast Cancer Is Correlated with Drug Resistance

Breast cancer (BC) is one the most common malignant tumors among women worldwide. Interestingly, miR-202 was found to be upregulated in most BC patient blood (Table 1) [9,10,11,13,14]. On the other hand, lower expression levels of miR-202 noted in BC cell lines or tissue samples [15,16,17]. A positive correlation has been observed between downregulation of miR-202 in BC tissues and cell lines and poor prognosis of BC patients [18]. This poor prognosis of patients with reduced miR-202 levels led to the proposition that miR-202 acts as tumor suppressor via regulating the oncogene KRAS thereby reducing proliferation and metastasis [15]. Higher levels of miR-202-5p in BC tissues have also been attributed to the resistance to drugs, where miR-202-5p overexpression was seen in both patients with doxorubicin (DOX)-resistant tumors and in DOX-treated MCF-7 BC cells [13]. Overexpression of miR-202-5p enhanced the proliferation and drug resistance of BC cells in vitro and in vivo [13]. Consistent with that, miR-202-5p was found to be upregulated in drug-resistant triple negative carcinoma, in BC tissues and cell lines [14]. In addition, the post-operative serum level of miR-202 2–3 months after surgery was lower than that in patients before operation, while high levels of miR-202 in patient sera significantly correlated with poor overall survival [10]. Thus, miR-202 appears to be upregulated in most breast tumors, which is not the case in most other types of cancers; this could be related to the resistance of BC cells to drugs, and probably to a difference between miR-202 expression in tissues and blood [13,14]. Therefore, miR-202 may be considered as a prognostic biomarker in BC where its upregulation positively correlates with the extent of drug resistance and the aggressiveness of the tumor.

## 4. miR-202 as a Novel Gastrointestinal Tract Tumor Suppressor

Numerous clinical studies have addressed the role of miR-202 in gastrointestinal tract tumors (GIT), including oral [12], esophageal [19,20,21], gastric [22,23,24], pancreatic [25,26,27,28], hepatocellular [29,30,31], and colorectal [32,33,34,35] cancers, and documented a lower expression of miR-202 in tumor tissues and a tumor suppressive function of miR-202 overexpression on GIT cancer progression. In oral cancer cell lines, overexpression of miR-202 downregulated the protein expression level of the transcription factor Sp1, which, in turn, reduced cancer cell migration and invasion. Inhibition of miR-202, however, markedly enhanced oral cancer progression [12], indicating a suppressor function of miR-202 in this tumor type. A tumor suppressive function for miR-202 has also been proposed by Meng and colleagues in esophageal squamous cell carcinoma [19,20,21]. miR-202 expression levels and involved downstream/upstream molecules in various types of gastrointestinal tract tumor are listed in Table 2.

### 4.1. Downregulation of miR-202 in Gastric Cancer Is a Potential Biomarker for Tumor Progression

Worldwide, gastric cancer (GC) is one of the most frequent causes of cancer mortality [37,38]. To date, three clinical studies investigated the role of miR-202 in GC and the obtained conclusive results revealed that miR-202 is downregulated in tumor tissues relative to the adjacent healthy tissues [22,23,24]. Interestingly, miR-202 expression levels were found to vary with the tumor size and patient age [22]. In addition, overexpression of miR-202-3p in GC cell lines caused a marked suppression of cell proliferation and induced apoptosis under in vitro condition and in xylographed nude mice. This tumor suppression activity occurred via direct targeting the transcription factor Gli1 and inhibition of the expression of the Gli1 target genes γ-catenin and BCL-2 [22]. Similarly, miR-202 was reported to be targeted by the LncRNA MALAT1, whose knockdown significantly reduced the expression of Gli2 via negative regulation of miR-202. In agreement, a negative correlation has recently been observed between miR-202-3p and MALAT1 expression, where upregulation of the latter increased the level of the splicing factor SRSF1 via targeting miR-202-3, thus activating the mTOR pathway to enhance GC migration and epithelial-mesenchymal transition (EMT) [24]. On the other hand, miR-202-3p was the most extraordinarily upregulated miRNA in type 1 gastric neuroendocrine neoplasm [4]. In all, downregulation of miR-202 in GC (Table 2) is a potential biomarker indicative of tumor progression.

### 4.2. The Role of miR-202 in Pancreatic Cancer and Hepatocellular Carcinoma

Pancreatic cancer (PC) is currently rated as the fourth leading cause of cancer-related death worldwide [39]. A tumor suppressor function for miR-202 in pancreatic carcinoma is suggested by all preclinical and clinical studies up to date [25,26,27,28]. lncRNA NORAD and ANP32E were upregulated in PC tissues and cells, whereas the miR-202-5p level was down-regulated. lncRNA NORAD competitively bound to and sequestered miR-202-5p which promoted the expression of the miR-202-5p target gene ANP32E that enhanced PC cell viability, proliferation, and self-renewal ability in vitro, as well as stimulating tumorigenesis of PC stem cells in vivo [28]. In stellate pancreatic cells, miR-202 overexpression slowed growth as well as reduced stromal extracellular membrane matrix protein expression. In orthotopic PC mouse models, both immunodeficient and immunocompetent, miR-202 overexpression reduced tumor burden, and metastasis [26]. While decreased miR-202 expression in PC tissues correlated with a poor prognosis of PC patients and an elevated cellular proliferative capacity [27], its overexpression in PC cells reduced cell proliferation and tumorigenesis by also impairing glycolysis [27]. These data suggest that overexpression of miR-202 is associated with tumor control in PC.

Hepatocellular carcinoma (HCC) is the third leading cause of cancer-related deaths worldwide [39]. In the five articles that to date investigated the clinical role of miR-202 in HCC, lower expression levels of miR-202 were found to be associated with increased tumor size, vascular invasion, progressed tumor node and metastasis stages, and poor overall survival rates [14,29,30,31,40]. Mechanistically, in cell lines and xenograft nude mouse models, miR-202 significantly inhibited HCC cell proliferation and EMT, induced apoptosis, and suppressed tumor formation. In a xenograft nude mouse model, it was shown that the binding of miR-202 to BCL2 mRNA downregulated the expression of this protein [29]. In addition, upregulation of miR-202 in vitro inhibited cell proliferation by regulating hexokinase 2 (HK2) expression in HCC [31]. Similarly, the low-density lipoprotein receptor-related protein 6 (LRP6) was demonstrated to be a direct target of miR-202 where the latter suppressed the expression of LRP6 by binding to the 3′-untranslated region (UTR) of its mRNA, while overexpression of miR-202 in HCC cells suppressed LRP6, reducing cell proliferation and tumorigenicity [29]. Therefore, the available results indicate that lower level of miR-202 in HCC cancer (Table 2) is a potential biomarker of tumor progression.

### 4.3. miR-202 in Colorectal Tumors

Colorectal cancer (CRC) is one of the most commonly diagnosed tumors in men and women worldwide [41,42]. However, several gene expression studies in the last decades documented downregulation of miR-202 in colorectal tumors of patients relative to the adjacent healthy tissues, suggesting a potential prognostic value for miR-202 in colorectal cancer [32,33,34,35]. Lower levels of miR-202-5p in CRC tissues was found to be positively correlated with postoperative survival, and overexpression reduced the proliferation rate and inhibited tumor growth and metastasis of CRC cells [32,33,34,35]. On the other hand, a more recent study showed miR-202-5p up-regulation in CRC tumors and that its over-expression was critical for CRC cell viability [36]. However, based on the available preclinical and clinical information (Table 2), a tumor suppressive role of miR-202 in colorectal tumorigenesis appears to predominate.

## 5. Tumor Suppression Function of miR-202 in Non-Small Cell Lung Cancer

Non-small cell lung cancer (NSCLC) is the predominant form of lung cancer and an aggressive disease, accounting for up to 85% of newly diagnosed lung cancer cases [43]. A tumor suppressor function for miR-202 in NSCLC has been documented by preclinical and clinical studies [34,44,45,46,47,48,49,50,51], except for one recent study that reported a higher level of circulating miR-202 prior to the start of first-line chemotherapy (Table 3) [8]. In a study on the roles of microRNAs in NSCLC metastasis, miR-202 occurred to be the down-regulated during the development of NSCLC metastasis [51] as was the case in asbestos-induced lung cancers [44]. Mechanistically, miR-202-3p was suggested to be an upstream negative regulator of the endopeptidase Matrix Metallopeptidase MMP-1, thereby inhibiting the proliferation, migration and invasion of lung adenocarcinoma cells [52]. Furthermore, Tiansheng and colleagues noted that a high expression of lncRNA Metastasis Associated Lung Adenocarcinoma Transcript 1 (MALAT1) was associated in NSCLC cancers with a lower expression of miR-202 and with large tumor size, advanced cancer, and tumor metastasis [49]. Similarly, the downregulation of miR-202 was associated with lymph node metastasis and progressed node metastasis stage, elevated STAT3 expression and proliferation [46]. Overexpression of miR-202, on the other hand, reduced NSCLC cell viability, migration, and invasion in vitro [46]. Mechanistic studies suggest that overexpression of miR-202 may target the Ras/mitogen-activated protein kinase (MAPK) pathway and thereby enhance the apoptosis signaling cascade induced by cisplatin in NSCLC cells [47]. miR-202 was also reported to act as a tumor suppressor through targeting cyclin D [45]. In agreement, a lower expression of miR-202 and high expression of KCNK15 and WISP2 antisense RNA 1 (KCNK15-AS1) in fresh lung adenocarcinoma samples was associated with poor prognosis, while silencing of KCNK15-AS1 inhibited lung cancer cell proliferation via upregulation of miR-370 and miR-202 [48]. In contrast, a higher level of miR-202 in patients’ plasma was found to be associated with disease progression and poor prognosis among patients through interfering with macrophage polarization in NSCLC patients [8]. Based on the available data from tissue samples and cell line studies, miR-202 appears to act as tumor suppressor in NSCLC too. In contrast, one report has suggested that higher expression of circulating miR-202 in blood is associated with poor prognoses [8]. However, this study used only blood samples and thus it remains to be determined whether there is an inverse correlation of blood miR-202 values and tumor expression levels, or whether the regulatory circuits of miR-202 are altered.

## 6. Tumor Suppressor Function of miR-202 in Different Reproductive System and Gynecological Cancers

miR-202 expression levels in different malignancies of the male and female reproductive system are listed in Table 4. Gynecologic cancers are tumors of the female reproductive organs, of which cervical, ovarian, uterine, and vaginal are the most common. To date, four studies have investigated miR-202 role in cervical cancer (CC), three of them suggest a tumor suppressor role of miR-202 in cervical cancer [53,54,55]. Mechanistically, miR-202 was found to inhibit CC cell proliferation, migration, and invasion via direct targeting of cyclin D1 [54]. In addition, overexpression of miR-202-5p suppressed the expression of PIK3CA and inhibited the activation of PI3K/Akt/mTOR signaling pathway, suppressing proliferation and the progression of epithelial-mesenchymal transition as well as the invasion of cervical cancer [55]. Furthermore, the axis of lncRNA MALAT1/miR-202-3p/matricellular proteins, periostin, was shown to play an important role in regulating cell viability, cell migration and invasion, and EMT of cervical cancer cells through activating Akt/mTOR signaling pathway, whereas miR-202-3p upregulation reduced cancer progression [53]. In contrast, in liquid biopsies of cervical intraepithelial neoplasia, a specific type of CC, miR-202-3p was identified among an upregulated set of nine miRNAs [56]. In fact, among the reproductive system malignancies, ovarian cancer (OC) is the third leading cause of death (American Cancer Society, 2021). MiR-202-5p expression was found to be reduced in OC and to be positively related to HOXB2, while its over-expression led to a higher 5-year survival rate [57]. Likewise, in rare gynecologic tumors, i.e., ovarian germ cell tumors (OGCTs) and sex cord stromal tumors (SCSTs), a lower expression of miR-202-3p was reproducibly observed in malignant OGCTs as compared to benign OGCTs or SCSTs tumors [58]. On the other hand, miR-202 was among 4 miRNAs that showed a higher expression in preoperative sera of ovarian cancer patients [6].

The role of miR-202 in prostate cancer has been investigated in two recent clinical studies [63,64]. According to Zhang and colleagues (2018), miR-202 is downregulated in human prostate cancer tissues and cell lines, while overexpression of miR-202 significantly suppressed prostate cancer progression, demonstrating miR-202 as a tumor suppressor via direct targeting of PIK3CA [63]. However, a role of miR-202 as tumor suppressor in prostate cancer is not obvious in the data of McDonald et al. [64]. Thus, more studies are needed to signify the role of miR-202 in PC.

Endometrial cancer (EC) is among the major cancers of the female reproductive system [65,66]. As shown in many other types of cancers, cancerous tissues of EC patients showed downregulation of miR-202, which was associated with poor prognosis [59,60,61,67], indicating a suppressive role for miR-202 in this cancer type.

## 7. miR-202’s Role in Osteosarcoma

Osteosarcoma (OS) is a malignant tumor of bone tissue with a high metastatic potential [68]. To date, out of five studies investigating the role of miR-202 in OS (Table 5), four have demonstrated a potential tumor suppressor function [69,70,71,72]. However, a study analyzing biopsies of eight early OS tumor tissues paired to their normal adjacent tissues, miR-202 appeared to be significantly upregulated in the tumor tissues, while miR-202 was found to play a role in chemoresistance by inhibiting apoptosis in TGF-β1-treated OS cell lines through targeting the suppressor gene *PDCD4* [73]. Tumors with high TGF-β expression correlate with poor prognosis with regard to chemoresistance [74]; it will, therefore, be of interest to investigate miR-202 expression levels in such a cohort. In a search for biomarkers of metastatic OS, downregulation of miR-202 was found to be pivotal for OS metastasis, and its expression was relevant for other genes involved in metastatic progression [71]. In agreement with that, human OS cell lines and tumor tissue samples displayed downregulated miR-202 expression, while restoration of miR-202 expression inhibited OS cell proliferation, enhanced apoptosis, and suppressed tumor growth in nude mouse models [69,72]. Reduced levels of miR-202 in tumor samples and in a metastasis in vivo model indicated that the downregulation of miR-202 expression was associated with lung metastasization of osteosarcoma, driven by the lncRNA MALAT1 sequestering miR-202 [72].

## 8. Tumor Suppression Role of miR-202 in Further Types of Cancer

A potential tumor suppression function for miR-202 has been documented in several other tumors (Table 6) including thyroid carcinoma [67,75], multiple myeloma [76,77,78,79], glioma [80], and follicular lymphoma [81].

In multiple myeloma cell lines and cancer tissue samples, a tumor suppression function for miR-202 was evident [76,77,78,79]. However, only one study pointed towards an oncogenic action of miR-202, as the relative expression of miR-202 in the serum of MM patients was significantly higher than that in healthy controls [5]. It will, therefore, be interesting to learn whether miR-202 levels in serum reflect the expression levels in corresponding MM tissues.

In the neuroplastoma cell line LAN-5, miR-202 was observed to be activated through E2F1, which in turn downregulated MYCN protein expression [88]. Since miR-202 was a strong negative regulator of *MYCN* expression, a strategy for neuroblastoma treatment further research has to reveal whether targeting of the MYCN axis by instigating the expression of miR-202 is an option for neuroblastoma treatment [87,88]. In this respect it may be of interest to test whether the Shikonin quinone, that can upregulate miR-202 and inhibit MYCN expression in retinoblastoma cells [90], may also exert such function in MM cells.

## 9. Mechanistic Pathways Involving miR-202 as a Tumor Suppressor

As outlined above, miR-202 was found to work as a tumor suppressor in various types of cancers. To conduct its tumor suppression function miR-202 is targeted by several signaling molecules and influences downstream effectors as indicated in Figure 1.

The genetic interaction of miR-202 in tumor suppression was addressed by association analysis in follicular lymphoma patients [91], which revealed a significant association between a germline mutation (rs12355840) in the miR-202 precursor sequence and follicular lymphoma (FL) risk. However, several recent studies [23,51,76,92] have shown that various signaling molecules/pathways may interfere with or modulate miR-202’s tumor suppression function. These include the long non-coding RNAs (lncRNA) MALATA and NORAD, the tumor suppressor, PTEN, and other signaling targets (Figure 1) as addressed below.

### 9.1. The lncRNA Metastasis-Associated Lung Adenocarcinoma Transcript 1 (MALAT1) Promotes Cancer Progression via Down-Regulating miR-202 Expression

Metastasis-associated lung adenocarcinoma transcript-1 (MALAT1) is a lncRNA that was initially found to be overexpressed in early NSCLC [91], where it can regulate downstream target molecules by directly binding to miRNAs and thereby enhancing cell proliferation, metastasis, and invasion, thus fueling cancer progression. miR-202 was reported to be a direct downstream target of MALAT1 in different types of tumors including NSCLC [49], osteosarcomas [72], gastric [23,24,92], and cervix carcinoma [53], where a negative correlation was found between the expressions of MALAT1 and miR-202. Based on structural analysis, it was proposed that the complementary sequence of miRNA in lncRNA acts as a competing endogenous RNA (ceRNA) that directly binds to miR-202 and sequesters it in NSCLC. At the clinical level, higher level of MALAT1 was reported to have an inhibitory effect on miR-202 which is related to large tumor size, poor histological grade, and tumor metastasis in NSCLC [49]. On the other hand, miR-202 overexpression inhibited cell proliferation and invasiveness of MALAT1-overexpressing cells [49]. In agreement, downregulation of MALAT1 or the upregulation of miR-202 was reported to reduce lung metastasis of osteosarcoma tumors [72]. MALAT1 was also found to be upregulated in gastrointestinal cancer tissues and its higher expression correlated with larger tumor size and lymph node metastasis [23]. In gastric carcinoma, there was also a negative correlation of MALAT1 and miR-202, with knockdown of MALAT1 restoring miR-202 expression and reducing transcription factor GLI2 expression, leading to growth inhibition [23]. In another study, the chemokine CCL21 was reported to induce MALAT1 expression that is functionally targeting miR-202-3p, while it upregulated the splicing factor SRSF1 to activate mTOR signaling pathway, thereby promoting the migration and EMT of GC cells [24]. Similarly, MALAT1 was found to enhance the proliferation ability of gastric cancer cells by inhibiting the expression of a set of miRNAs including miR-202 [92]. In agreement, the expression levels of the extracellular protein periostin and MALAT1 were found to be negativity correlated with the expression of miR-202-3p in cervix carcinoma tissues. The MALAT1/miR-202-3p/periostin axis was associated with tumorous growth characteristics and EMT of CC cells through activation of AKT/mTOR signaling [53]. All these studies suggest that MALAT1 promotes tumor progression by the direct sequestration of the miR-202 RNA.

### 9.2. The lncRNA NORAD Inhibits miR-202 to Promote Cancer Progression

The “long non-coding RNA activated by DNA damage” (NORAD) is highly conserved and copious. Recent work highlighted an oncogenic function of NORAD [28,35,75]. An inverse relationship between NORAD and miR-202-5p expression was found to play a pivotal role for colorectal cancer progression, expressed by NORAD dependent downregulation of miR-202 expression in CRC tissues and xenografted cell lines [35]. Overexpression of NORAD also suppressed miR-202-5p and promoted EMT progression in thyroid carcinoma cells [75]. In pancreatic cancer stem cells (PCSCs), NORAD sequestered miR-202-5p and thereby promoted the expression of the miR-202-5p target gene *ANP32E*, which enhanced cell proliferation in vitro, as well as facilitating tumorigenesis of PCSCs in vivo [28]. NORAD overexpression was noted to promote HCC cell migration and invasion. In vivo, HCC tissues had a high level of NORAD compared with paratumor tissue and NORAD upregulation was associated with the shorter overall survival of patients with HCC [30]. Mechanically, NORAD might function as a competing endogenous RNA to regulate miR-202-5p, which acts as a tumor-suppressor via the TGF-β pathway. Thus, NORAD has a tumor-promoting effect in HCC, likely through sequestering miR-202. NORAD binding miR-202-5p has been shown to increase the expression of the multidrug resistance ABC transporter ABCB1, a target of miR-202-5p. Knockdown of NORAD reduced ABC transporter expression as did overexpression of miR-202-5p in A549/DDP cells [50]. This shows an important regulatory interaction between NORAD and miR-202.

### 9.3. miR-202 Suppresses Cancer via Targeting the Oncogene Metadherin and the Ras GTPase

The oncogene metadherin (MTDH), first identified bioinformatically as a direct target gene of miR-202 in glioma, was found to be upregulated and negatively correlated with miR-202 expression in clinical glioma tissues, and targeting of MTDH by miR-202 was reported to inhibit glioma cell proliferation, migration, and invasion via impairment of the PI3K/AKT and WNT/β-catenin pathways [80]. On the other hand, in NSCLC cells, overexpression of miR-202 was found to inhibit the Ras/mitogen activated protein kinase (MAPK) pathway via targeting the *KRAS* gene, thereby expanding apoptosis signaling induced by cisplatin [47]. This aligns with the observation that endometrial stromal cell migration and invasion is suppressed by the K-Ras/Raf1/MEK/ERK signaling pathway [93]. Additionally, ionizing radiation (IR) was reported to activate K-Ras/ERK signaling downregulated miR-202 and miR-185 expression, which, in turn, upregulated CD44. The latter was then shown to induce cancer stemness and EMT features of grade IV gliomas [86]. These studies show that miR-202 is important to control oncogene expression in signaling pathways during carcinogenesis (see below).

### 9.4. Cyclin D1 Is a Downstream Target of miR-202

In cervical and lung cancers, miR-202 was found to play a crucial role in suppressing cell proliferation and tumor progression by directly targeting cyclin D1 [45,48,54]. Interestingly, in human Sertoli cells, it has been shown that miR-202-3p controls apoptosis, proliferation, and synthesis functions via targeting LRP6 and Cyclin D1 of the Wnt/β-catenin signaling pathway [30].

### 9.5. miR-202 Binds to B Cell-Activating Factor (BAFF) and Enhances Tumor Suppression

A potential binding site for miR-202 in the B cell-activating factor (BAFF) RNA was indicated by bioinformatic analysis and confirmed by in vitro molecular assays [77]. Thus, miR-202 can inhibit malignant myeloma cell proliferation and induce apoptosis via regulating BAFF and targeting the JNK/SAPK signaling pathway [76,77,79]. Similar observations were made in U266 cells [76]. In line with that, upregulation of miR-202 in the presence of the drug Bortezomib inhibited MM cell survival more effectively compared to Bortezomib treatment alone, which involved the reactivation of the JNK/SAPK signaling pathway as the regulatory downstream target of miR-202 [79]. miR-202 also regulated the expression of BAFF in bone marrow stromal cells, where over-expression of miR-202 in BMSCs rendered them more sensitive to Bortezomib; the JNK/SAPK signaling pathway was involved in the regulatory effect of miR-202 on drug resistance of MM cells which was attributed to the inhibitory effect of miR-202 on BAFF expression and downregulation of the JNK/SAPK signaling pathway [79].

## 10. The miR-202/PTEN/AKT Axis in Tumor Progression

In recent years, mechanistic studies linked miR-202 to the expression of the critical tumor suppressor phosphatase and tensin homologue deleted on chromosome 10 (PTEN). However, this interaction may have a tumor promoting potential as outlined below. In breast tumors, a direct interaction between PTEN by miR-202 has been observed [13,17]. Upregulation of miR-202-5p suppressed PTEN and promoted DOX resistance and cell proliferation as well as inhibiting apoptosis of MCF-7 breast cancer cells. In contrast, downregulated miR-202-5p induced the converse effects in MCF-7/DOX cells promoting the inhibition of the PI3K/AKT pathway [13]. Similar oncogenic functions for miR-202 were observed in human colorectal cancer samples that showed c-Myc driven upregulation of miR-202-5p suppressed PTEN, which in turn increased the phosphorylation of AKT and enhanced CRC cell proliferation [36]. In addition, we have recently shown that inhibiting the direct interaction between miR-202 and PTEN negatively regulated the PI3K/AKT pathway and thereby reduced the migration and the invasion of highly metastatic breast cancer cells [17]. Along this line, overexpression of miR-202-5p in cervical cancer reduced *PIK3CA* gene expression as well as the activation of PI3K/AKT/mTOR signaling pathway, which suppressed proliferation, invasion, and EMT [55].

## 11. Micro-RNA-Based Therapeutic Approaches

RNA-based drugs have attracted attention as possible promising therapeutics for treatment of cancer and other diseases. In this context, recent progress in the clinical application of various RNA types, including miRNAs, have attracted increased attention, not least because RNA-based vaccine development has been a success (see, e.g., [94,95]). miRNAs whose dysregulation is associated with cancer represent attractive targets for treatment development. Based on preclinical research, miRNA mimics and anti-miRNAs have led to promising developments, with several miRNA-targeted therapeutics reaching clinical trials. For example, a mimic of the tumor suppressor miR-34 has reached phase I clinical trials for cancer treatment [96]. To our knowledge, clinical trials implementing miR-202 have not yet been launched. However, it seems conceivable that the targeting of the MALAT1/NORAD axis converging on miR-202 or miR-202 inhibition in breast cancers may be of future interest.

## 12. Conclusions

Most of the studies reviewed above show that miR-202 is a regulatory micro-RNA whose normal expression puts a break on tumor development. However, this tumor suppression function of miR-202 can be perturbed by the upstream regulatory circuitry involving the production of the lncRNA MALAT1 or other upstream regulators/oncogenes, which varies according to tissue type and eventually tumor stage and type. In many cases downregulation of miR-202 fuels tumor development and metastasis, suggesting that miR-202 in most contexts acts as a tumor suppressor and that inhibition of its upstream down-regulatory elements can have suppressor effects. The downstream actions of miR-202 converge on the PI3K/AKT and Wnt/β-catenin pathways suggesting that modulation of miR-202 expression could be exploited in cancer therapy. Along this line, it is of interest that miR-202 can downregulate the oncogene metadherin in gliomas [80]. Furthermore, the expression levels of miR-202 in blood or serum may be used as a biomarker for disease progression. However, more conclusive studies are needed on the extent to which blood levels do realistically reflect expression levels in cancer tissues during tumor progression. Furthermore, the differential modulation of the expression of mir-202 in different tumor types suggests that the various outcomes of miR-202’s regulatory effects depend on the context of rewired regulatory circuits in tumor tissues and that a complex upstream regulation of miR-202 may lie behind its variable roles in tumorigenesis by modulating its downstream actions. In the worst case, upregulated miR-202 expression may rather be a side effect of a regulatory network installed by a cancer genome, which may deprive miR-202 of its ability to exert its tumor suppressor function that may still be fully evident in other contexts and so far in most tumors investigated.

## Figures and Tables

**Figure 1 ijms-23-05870-f001:**
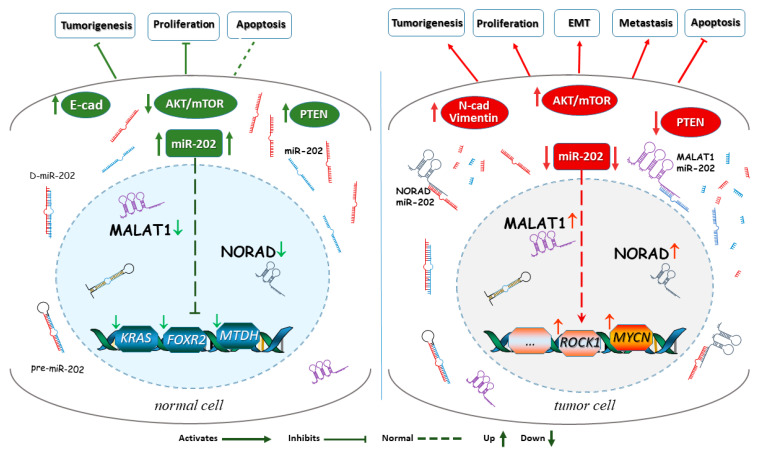
**Downstream and upstream signaling involved in miR-202’s suppressor functions**. A simplified scheme of the complex circuitry around miR-202 in normal and in cancer cells. In a normal cell (**left** detail), the lncRNAs like MALAT1 and NORAD and oncogenes display low expression levels, allowing higher levels of miR-202, which prevents an increased expression of tumor promoting genes, such as *KRAS*, *FOXR2*, and *MTDH*, and allows miR-202 to act on other downstream target molecules. miR-202 is also important for normal activation and phosphorylation events in the AKT pathway. In the cancer cell (**right** detail), the lncRNAs MALAT1 and NORAD as well as oncogenes reduce miR-202 expression leading to miR-202’s degradation and upregulation of cancer-promoting pathways as well as downregulation of apoptosis, which fuels tumor progression.

**Table 1 ijms-23-05870-t001:** Involvement of miR-202 in Breast Cancer progression and suppression.

mir-202 Function	Regulation (Up/Down)	Patients’ Samples, Cell Lines, Animal Model	Involved Downstream/Upstream Molecules	Ref.
pro-tumorigenic	up	23 DOX-resistant, 39 DOX-sensitive BC cancer samples, cells lines, xenografted nude mice	PTEN/PI3k/Akt	[13]
pro-tumorigenic	up	BC cell lines	PTEN/PI3k/Akt	[17]
pro-tumorigenic	up	45 pairs of BC cancer and adjacent normal tissue, cell lines	lncRNA GSEC, AXL	[14]
pro-tumorigenic	up (early stage)	Plasma samples of 30 BC patients (stages I–III), 30 control samples	n.d.	[11]
pro-tumorigenic	up	Sera of 102 BC patients and 26 with benign breast diseases, blood samples of 37 healthy controls	n.d.	[10]
pro-tumorigenic	up (early stages)	Blood samples of 48 early-stage BC and 57 controls	n.d.	[9]
suppressor	down	27 BC tissue samples, cell lines	ROCK1, E-cadherin, Twist, N-cadherin, and MMP2	[18]
suppressor	down	BC cell lines	MMP-1, claudin-5, ZO-1 and ß-catenin	[16]
suppressor	down	30 BC tissues	KRAS	[15]

n.d. = no data.

**Table 2 ijms-23-05870-t002:** The suppressive function of miR-202 in different types of digestive tract cancers.

mir-202 Function	Regulation	Samples, Cell Lines, Patient Material, Animal Model	Involved Downstream/Upstream Targets	Ref.
	(Up/Down)			
**Oral Cancer**				
suppressor	down	73 oral cancer tissue, 48 normal tissues, blood samples, cell lines	Sp1, protein kinase B	[12]
**Esophageal squamous cell carcinoma (ESCC)**				
suppressor	down	cell lines	HSF2/Hsp70	[20]
	down	76 esophageal cancers (44 ESCC, 32 EAC) with adjacent normal tissues	n.d.	[21]
suppressor	down	30 primary ESCC tissues and adjacent noncancerous tissues, cell lines	p-FAK, p-Akt, LAMA1	[19]
**Gastric Cancer**				
suppressor	down	60 GC tissues and adjacent normal tissues	MALAT1	[23]
suppressor	down	150 GC tissues and adjacent normal tissues, mouse xenografts	Gli1, γ-catenin, BCL-2	[22]
suppressor	down	115 GC tissues with normal tissue samples, cell lines, mouse xenografts	MALAT1, SRSF1, CCL21, mTOR	[24]
**Hepatocellular carcinoma**				
suppressor	down	56 HCC samples, cell lines	hexokinase 2, glycolysis	[31]
suppressor	down	Eight pairs of snap-frozen HCC tumor, cell lines	LRP6	[29]
suppressor	down	Tissues from 95 patients with HCC who underwent tumor, cell lines, Tumor xenograft	NORAD, NORAD/miR-202-5p/TGF-β axis	[30]
**Gall bladder cancer**				
pro-tumorigenic	up	Blood samples of 40 GBC patients	n.d.	[7]
**Pancreatic cancer**				
suppressor	down	PC cell lines & orthotopic mouse xenografts	TGFβ1, EMT	[26]
suppressor	down	cell lines	hexokinase 2, inhibited glycolysis	[27]
suppressor	down	cell lines	*Mxd1, SAP18*	[25]
suppressor	down	28 PC cancer and adjacent normal tissue samples, cell lines, mouse xenografts	lncRNA NORAD and ANP32E	[28]
**Colorectal cancer**				
suppressor	down	68 cases of CRC and 66 adjacent normal tissues, cell lines and xenografted nude mice	UHRF1	[33]
suppressor	down	68 CRC tissues, cell lines	1SMARCC1	[32]
suppressor	down	47 pairs of CRC tissues and adjacent normal tissues, cell lines and xenografted nude mice	NORAD	[35]
suppressor	down	98 primary CRC cancer tissues and adjacent normal tissues, cell lines, mouse xenografts	ADP-ribosylation factor-like 5A	[34]
pro-tumorigenic	up	25 CRC tumors and adjacent normal tissues, cell lines	PTEN, c-Myc AKT	[36]

n.d. = no data.

**Table 3 ijms-23-05870-t003:** miR-202 is a potential tumor suppressor in lung cancer.

mir-202 Function	Regulation	Samples, Cell Lines, Patient Material, Animal Model	Involved Downstream/Upstream Molecules	References
suppressor	down	56 patient tissues and corresponding normal tissues, cell lines	STAT3	[46]
suppressor	down	40 lung adenocarcinoma and adjacent normal tissue samples	KCNK15-AS1, EGFR, miR-202/miR-370/EGFR axis	[48]
suppressor	down	40 NSCLC tissues, cell lines	MALAT1	[49]
suppressor	down	25 NSCLC and adjacent paratumor tissues, cell lines, mouse xenografts	KRAS	[47]
suppressor	down	cell lines	NORAD	[50]
suppressor	down	20 lung cancer and adjacent normal tissues	cyclin D1	[45]
suppressor	down	26 tumor and corresponding normal lung tissue samples	n.d.	[44]
suppressor	down	Nude mice, cell lines		[34]
pro-tumorigenic	up	125 plasma samples first-line chemotherapy	Macrophage Polarization	[8]

n.d. = no data.

**Table 4 ijms-23-05870-t004:** Reproductive system and gynecological cancers and role of miR-202.

miR-202 Function	Regulation	Samples, Cell Lines, Patient Material, Animal Model	Involved Downstream/Upstream Molecules	Ref.
**Cervical cancer**				
suppressor	down	105 patient samples, cell lines	PIK3CA/PI3K/Akt/mTOR pathway	[55]
suppressor	down	100 CC and control tissues, cell lines	cyclin D1	[54]
suppressor	down	23 CC tissues, cell lines	MALAT1/miR-202-3p/periostin, EMT, Akt/mTOR signaling	[53]
**Endometrial cancer**				
suppressor	down	90 tumor & 40 corresponding. normal tissues, cell lines, xenografted nude mice	*FOXR2* oncogene	[59]
suppressor	down	76 EC tissue samples	FGF2 and Wnt/β-catenin	[60]
suppressor	down	20 EC and adjacent normal tissue samples, cell lines	NEAT1, TIMD4	[61]
**Uterine Leiomyosarcoma**				
pro-tumorigenic	up	39 paraffin UL tumor samples, cell lines	MYCN	[62]
**Ovarian cancer**				
pro-tumorigenic	up	21 patient serum samples	n.d.	[6]
suppressor	down	55 paired OC and para-tumor tissues, cell lines	HOXB2	[57]
pro-tumorigenic	up	23 non-epithelial ovarian germ cell tumors, 16 ovarian, 7 sex cord stromal tumors	n.d.	[58]
**Prostate cancer**				
suppressor	down	54 PC tissues and 11 adjacent normal prostate tissues, cell lines and mouse xenografts	PIK3CA	[63]
inconclusive	up in high grade PC	66 sera of patients with low- and high-grade PC	n.d.	[64]

n.d.= no data.

**Table 5 ijms-23-05870-t005:** Role of miR-202 in Osteosarcoma.

mir-202 Function	Regulation	Samples, Cell Lines, Patient Material, Animal Model	Involved Downstream/Upstream Molecules	Ref.
suppressor	down	36 tumor specimens and normal adjacent tissues, cell lines	ROCK1	[70]
pro-tumorigenic	up	8 paired tumor and normal adjacent tissues, cell lines	PDCD4	[73]
suppressor	down	16 tumor tissues and matched adjacent normal tissues, cell lines	Gli2	[69]
suppressor	down	6 metastatic OS and non-metastatic OS	CALD1, STX1A	[71]
suppressor	down	32 OS without metastasis and 24 w lung metastases, 30 controls, cell lines	MALAT1	[72]

**Table 6 ijms-23-05870-t006:** miR-202 as a tumor suppressor in other types of cancer.

mir-202 Function	Regulation	Samples, Cell Lines, Patient Material, Animal Model	Involved Downstream/Upstream Molecules	Ref.
**Chronic myeloid leukemia**				
pro-tumorigenic	up	30 patient samples, cells lines, xenografts of nude mice	STAT5A/miR-202-5p/USP15/Caspase-6 regulatory axis	[82]
suppressor	down	15 CML samples, cell lines	HK2	[83]
**Follicular lymphoma**				
suppressor	down	cell lines	HAS2, FAM135A	[81]
**Glioma**				
suppressor	down	43 glioma samples and adjacent normal tissues, cell lines	MTDH, PI3K/Akt, Wnt/β-catenin pathways	[80]
**Laryngeal cancer**				
pro-tumorigenic	up	48 laryngeal primary tumors	n.d.	[84]
**Multiple myeloma**				
pro-tumorigenic	up	cell lines	BAFF, JNK/SAPK signaling pathway	[76]
pro-tumorigenic	up	40 MM patient sera, 30 healthy controls	BAFF	[5]
suppressor	down	cell lines	BAFF, JNK/SAPK signaling pathway	[77,79]
suppressor	down	7 MM patient samples, cell lines	BAFF	[85]
**Thyroid carcinoma**				
suppressor	down	40 PTC and adjacent normal tissues	lncRNA NORAD, EMT	[75]
suppressor	down	96 pairs of PTC and adjacent normal tissue, cell lines	β-catenin, WNT signaling	[67]
**Glioblastoma multiforme**				
suppressor	down	cell lines	IR-induced K-RAS/ERK signaling, CD44	[86]
**Neuroblastoma**				
suppressor	down	cell lines	MYCN	[87]
suppressor	down	cell lines	MYKN, E2F1	[88]
**Urinary Bladder cancer**				
suppressor	down	50 cancer and corresponding tissues, cell lines	EGFR	[89]

n.d. = no data. PTC: papillary thyroid carcinoma.

## Data Availability

Not applicable.

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
