# Peer review of "The microRNA-202 as a Diagnostic Biomarker and a Potential Tumor Suppressor"

_ijms, 2022, doi:10.3390/ijms23115870_

Round 1

Reviewer 1 Report

The study data are very comprehensive and valuable. The main remark is that there are many mistakes in writing. English spelling and grammar should be checked all over the manuscript. Additionally, you find that miR-202 appears to be upregulated in most breast tumours, which is not the case in other types of cancers, where it has mainly tumour suppressive function. You assume that this could be related to the resistance of breast cancer cells to drugs, and probably to the difference between miR-202 expression in tissues and blood. This latest statement should be explained somehow in the manuscript. The other minor remarks are indicated below.

The Title:

In the title there is apparently mistake and all over the manuscript it should be specified if it will be writtenMicro-RNA-202” or “microRNA” or “miRNA” or “miR-202”…

Abstract:

In this part and all over the manuscript, the word “signaling” should be corrected to “signalling”.

  1. MiR-202 as a novel gastrointestinal tract tumor suppressor

The sentence “MiR-202 expression level in various types of gastrointestinal tract tumor (Table 2)”…. is missing a part.

Additionally, “miR-202” is not written uniformly across the manuscript.

  1. Tumor suppressor function of miR-202 in different genitourinary cancers

Table 4: there is different font size and/or style across the table.

  1. Mechanistic pathways involving miR-202 as a tumor suppressor

Figure 1. should be in better resolution.

  1. Conclusions

The finding that miR-202 appears to be upregulated in most breast tumours, while it is not the case in other types of cancers, should be clearly discussed in this chapter.

Author Response

Reviewer 1:

The study data are very comprehensive and valuable. The main remark is that there are many mistakes in writing. English spelling and grammar should be checked all over the manuscript. Additionally, you find that miR-202 appears to be upregulated in most breast tumours, which is not the case in other types of cancers, where it has mainly tumour suppressive function. You assume that this could be related to the resistance of breast cancer cells to drugs, and probably to the difference between miR-202 expression in tissues and blood. This latest statement should be explained somehow in the manuscript. The other minor remarks are indicated below.

The Title:

In the title there is apparently mistake and all over the manuscript it should be specified if it will be written “Micro-RNA-202” or “microRNA” or “miRNA” or “miR-202”…

These typing errors have been corrected and throughout the manuscript and the writing unified.

Abstract:

In this part and all over the manuscript, the word “signaling” should be corrected to “signalling”.

Now the word "Signaling" has been corrected to "signalling"

  1. MiR-202 as a novel gastrointestinal tract tumor suppressor

The sentence “MiR-202 expression level in various types of gastrointestinal tract tumor (Table 2)”…. is missing a part.

Thanks for the hint. This sentence has been completed.

Additionally, “miR-202” is not written uniformly across the manuscript.

miR-202 is now written uniformly across the manuscript.

  1. Tumor suppressor function of miR-202 in different genitourinary cancers

Table 4: there is different font size and/or style across the table.

The font type/size has been adjusted to uniform size throughout the manuscript.

  1. Mechanistic pathways involving miR-202 as a tumor suppressor

Figure 1. should be in better resolution.

An updated figure with better resolution has been inserted.

  1. Conclusions

The finding that miR-202 appears to be upregulated in most breast tumours, while it is not the case in other types of cancers, should be clearly discussed in this chapter.

This point is now more clearly discussed, thank you for valuable comments.

Reviewer 2 Report

The review by Ahmed et al., entitled "The micrRNA-202 as a diagnostic biomarker and a potential tumor suppressor" deals with the role of miR-202 in several tumor entities. Overall it is well structured and cites a lot of recent studies on pro- as well as anti-tumorigeneic properties of this miRNA. To my best knowledge, no such review exists in the literature until today. 

I have some comments:

1) Line 53: "Upregulated miR-202 levels in patient blood has been suggested to be a novel bi- 53 omarker for early detection of different types of cancer including multiple melanoma 54 (MM) [5]" --> this should be myeloma

2) Table 1 header line: miR instead of Mir

3) Line 143 and line 158: unformatted reference "(Global Cancer Observatory, 2021)", alternatively use GLOBOCON reference by Sung et al. 2021, doi: 10.3322/caac.21660

4) Line 223: You refer to genitourinary cancer, but this definition would usually not include cervical cancer. I would suggest to either combine it with the breast cancer section or to split into urinary tract (bladder and urinary tract) and the more commonly used term gynecological malignancies or reproductive system, as you already used. May consider adding a germ line cancer subsection. 

5) Table 2: Please correct Prostate caner

6) Line 425: "Similarly, up- 425 regulated miR-202-5p in cervical cancer reduced PIK3CA gene expression and promoted the activation of PI3K/AKT/mTOR signaling pathway that suppressed the proliferation, invasion, and EMT [83]". This sound contradictory, as reduced PIK3CA expression would rather result in an inactivation instead of an activation of the PI3K/AKT/mTOR pathway as you wrote. Please check if this is correct. 

7) Figure 2: The Figure is somehow confusion in the upper part of the panels where the boxes and arrows are shown. The authors should consider using individual arrows to show the effects. If I understand correctly, the right panel displays a cancer cell, in which more degradation of miR-202 is occurring, ultimately leading to less inhibition of AKT/mTOR. However, this would then lead to a more active PI3K/AKT/mTOR and more downstream signaling and therefore increased proliferation etc. (as you stated). I was wondering why you choose a red downregulated arrow on AKT/mTOR in your Figure, as the activity would increase (doi 10.1023/B:APPT.0000045801.15585.dd).

8) The title should be changed to "miR-202 or microRNA-202" to match the manuscript.

9) The tables could be improved by using pro-tumorigeneic and anti-tumorigeneic and be more detailed regarding the subjects of the cited studies (column 3).

The authors piled up a lot of literature (which I really appreciate) but I think that this review could benefits from a more in depth analysis and conclusion of all the cited literature. It would significantly improve the manuscript if more outlook or potential applications of miR would be discussed. Topics that may be covered include (but are not limited to): Potential hurdles of the use of miR-based treatment, RNA-based medicine in general, biological properties of miRNAs.

Additional remarks:

  • Genes should be written in italic, proteins in non-italic font. Please check carefully throughout the entire manuscript what you are referring to.
  • Some abbreviations were not mentioned before used. The authors should check if all abbreviations have been defined. 
  • Some fonts are not black (e.g. Table 6, ref 74 metadherin).
  • The manuscript should be checked and revised by a native speaker.
  • The authors should extensively check the manuscript for typos before submitting the revision. 
  • Please add a short methods section in which the search terms, strategy (e.g., databases, language, meSH terms, year of publication) and inclusion criteria are stated.
  • Figure 2: The Figure has a really low quality. Please provide a higher resolution. 

Overall, I think that the work is solid, but still lacks interpretation. I would suggest a revision to give the authors the opportunity to improve their manuscript. 

Author Response

Reviewer 2

The review by Ahmed et al., entitled "The micrRNA-202 as a diagnostic biomarker and a potential tumor suppressor" deals with the role of miR-202 in several tumor entities. Overall it is well structured and cites a lot of recent studies on pro- as well as anti-tumorigeneic properties of this miRNA. To my best knowledge, no such review exists in the literature until today. 

I have some comments:

  • Line 53: "Upregulated miR-202 levels in patient blood has been suggested to be a novel bi- 53 omarker for early detection of different types of cancer including multiple melanoma54 (MM) [5]" --> this should be myeloma

Thanks for the hint, this error has been corrected.

  • Table 1 header line: miR instead of Mir

miR-202 is now written uniformly throughout the manuscript.

  • Line 143 and line 158: unformatted reference "(Global Cancer Observatory, 2021)", alternatively use GLOBOCON reference by Sung et al. 2021, doi: 10.3322/caac.21660

GLOBOCAN 2021 reference has been added as suggested.

  • Line 223: You refer to genitourinary cancer, but this definition would usually not include cervical cancer. I would suggest to either combine it with the breast cancer section or to split into urinary tract (bladder and urinary tract) and the more commonly used term gynecological malignancies or reproductive system, as you already used. May consider adding a germ line cancer subsection.

Thanks for this hint. We moved the urinary bladder cancer part to the “other types of cancer” table 6, and both the reproductive and gynecological cancer parts have been combined in one table (now table 4).  

5) Table 2: Please correct Prostate caner

This typing error has been corrected.

6) Line 425: "Similarly, up- 425 regulated miR-202-5p in cervical cancer reduced PIK3CA gene expression and promoted the activation of PI3K/AKT/mTOR signaling pathway that suppressed the proliferation, invasion, and EMT [83]". This sound contradictory, as reduced PIK3CA expression would rather result in an inactivation instead of an activation of the PI3K/AKT/mTOR pathway as you wrote. Please check if this is correct. 

Sorry, you are right. We rephrased to correctly state that overexpression of miR-202-5p suppressed the expression of PIK3CA and inhibited the activation of PI3K/Akt/mTOR signaling pathway.

7) Figure 2: The Figure is somehow confusion in the upper part of the panels where the boxes and arrows are shown. The authors should consider using individual arrows to show the effects.

Figure 2, has been now updated.

If I understand correctly, the right panel displays a cancer cell, in which more degradation of miR-202 is occurring, ultimately leading to less inhibition of AKT/mTOR. However, this would then lead to a more active PI3K/AKT/mTOR and more downstream signaling and therefore increased proliferation etc. (as you stated). I was wondering why you choose a red downregulated arrow on AKT/mTOR in your Figure, as the activity would increase (doi 10.1023/B:APPT.0000045801.15585.dd).

Thanks, you are right, the right part is the cancer cell (now stated as such below) and the AKT/mTOR pathway should be up. The red down arrow was a mistake. We have corrected and reworked Fig. 1 and replaced the figure with a new high resolution tiff file.

8) The title should be changed to "miR-202 or microRNA-202" to match the manuscript.

In the title we now write out and in text follow uniformly miR-202 throughout the manuscript.

9) The tables could be improved by using pro-tumorigeneic and anti-tumorigeneic and be more detailed regarding the subjects of the cited studies (column 3).

Good point. In the tables we have replaced “enhanced cancer progression” with “pro-tumorigenic” and specified the subjects used in the study of the column 3. We, for didactic reasons, like to remain with the expression “suppressor”, as it distinguishes more easily from ‘pro-tumorigenic’ which in narrow table text looks pretty similar to ‘anti-tumorigenic’.

The authors piled up a lot of literature (which I really appreciate) but I think that this review could benefits from a more in depth analysis and conclusion of all the cited literature. It would significantly improve the manuscript if more outlook or potential applications of miR would be discussed. Topics that may be covered include (but are not limited to): Potential hurdles of the use of miR-based treatment, RNA-based medicine in general, biological properties of miRNAs.

We have addressed this point and added a new section 11 on “micro-RNA-based therapeutic approaches” where we also provide reviews with further insights to these approaches and their current state.

Additional remarks:

  • Genes should be written in italic, proteins in non-italic font. Please check carefully throughout the entire manuscript what you are referring to.
  • This has been checked and changed where appropriate.
  • Some abbreviations were not mentioned before used. The authors should check if all abbreviations have been defined. 
  • This has been checked and non-mentioned abbreviation are now explained.
  • Some fonts are not black (e.g. Table 6, ref 74 metadherin).

The font type/size has been uniformly adjusted to black across the manuscript.

  • The manuscript should be checked and revised by a native speaker.

The manuscript language has been extensively revised according to the recommendations of a native speaker.

  • The authors should extensively check the manuscript for typos before submitting the revision.

That has been done and typos corrected.

  • Please add a short methods section in which the search terms, strategy (e.g., databases, language, meSH terms, year of publication) and inclusion criteria are stated.

This information has been inserted in the text at line 49.

  • Figure 2: The Figure has a really low quality. Please provide a higher resolution. 

We have redrawn the figure and updated with a high resolution version.

Overall, I think that the work is solid, but still lacks interpretation. I would suggest a revision to give the authors the opportunity to improve their manuscript. 

Thank you for this possibility and your constructive comments which have helped to improve our review

Round 2

Reviewer 2 Report

Dear authors, thank you very much for providing a revised form of the manuscript. 

I think the manuscript substantially improved due to the revisions.

I have no further comments.